# Artemisinin-Type Drugs in Tumor Cell Death: Mechanisms, Combination Treatment with Biologics and Nanoparticle Delivery

**DOI:** 10.3390/pharmaceutics14020395

**Published:** 2022-02-10

**Authors:** Xinyu Zhou, Fengzhi Suo, Kristina Haslinger, Wim J. Quax

**Affiliations:** Department of Chemical and Pharmaceutical Biology, Groningen Research Institute of Pharmacy, University of Groningen, Antonius Deusinglaan 1, 9713 AV Groningen, The Netherlands; xinyu.zhou@rug.nl (X.Z.); f.suo@rug.nl (F.S.)

**Keywords:** artemisinin, regulated cell death, combination treatment, nanoparticle delivery

## Abstract

Artemisinin, the most famous anti-malaria drug initially extracted from *Artemisia annua* L., also exhibits anti-tumor properties in vivo and in vitro. To improve its solubility and bioavailability, multiple derivatives have been synthesized. However, to reveal the anti-tumor mechanism and improve the efficacy of these artemisinin-type drugs, studies have been conducted in recent years. In this review, we first provide an overview of the effect of artemisinin-type drugs on the regulated cell death pathways, which may uncover novel therapeutic approaches. Then, to overcome the shortcomings of artemisinin-type drugs, we summarize the recent advances in two different therapeutic approaches, namely the combination therapy with biologics influencing regulated cell death, and the use of nanocarriers as drug delivery systems. For the former approach, we discuss the superiority of combination treatments compared to monotherapy in tumor cells based on their effects on regulated cell death. For the latter approach, we give a systematic overview of nanocarrier design principles used to deliver artemisinin-type drugs, including inorganic-based nanoparticles, liposomes, micelles, polymer-based nanoparticles, carbon-based nanoparticles, nanostructured lipid carriers and niosomes. Both approaches have yielded promising findings in vitro and in vivo, providing a strong scientific basis for further study and upcoming clinical trials.

## 1. Introduction

Nowadays, artemisinin (ART) and its derivatives offer the best protection against malaria. ART is a non-nitrogenous sesquiterpene lactone with a molecular formula of C_15_H_22_O_5_, which is derived from the Chinese plant *Artemisia annua* L. (also called sweet wormwood) [1,2]. The earliest record of using *Artemisia annua* L. as a drug (for treating hemorrhoids) was in a book named *Wushi’er Bingfang* (Prescriptions for Fifty-two Diseases) which was discovered in the Mawangdui tomb and was written around 215 BCE or even earlier [3,4]. Afterward, the plant was described as a malaria treatment in *Zhou Hou Bei Ji Fang* (Handbook of Prescriptions for Emergency, 326-341 CE) and *Dan Xi Xin Fa* (Danxi’s Mastery of Medicine, 1347 CE) [5,6]. In 1967, as an international help for Vietnamese soldiers suffering from malaria during the Vietnam War, “Project 523” was initiated by the Peoples’ Republic of China to research promising anti-malarial drugs [7]. Among all screened traditional Chinese medicines, realgar and artemisinin were two potential candidates with a 100% inhibition rate to rodent malaria [8]. Due to the toxicity of realgar in clinical trials, research was focused on the extraction and structural optimization of artemisinin [9,10].

The endoperoxide 1, 2, 4-trioxane bridge of ART was firstly identified by the Shanghai Institute of Organic Chemistry in 1975 [11,12]. It is cleaved by intracellular ferrous iron (Fe^2+^) into reactive oxygen species (ROS), resulting in the death of parasites. More ART derivatives were synthesized with the inherent endoperoxide bridge and a lactone ring, exhibiting better solubility and potency [13]. Dihydroartemisinin (DHA) was the first artemisinin derivative synthesized through the sodium borohydride reduction of ART [14]. It also acts as an intermediate in the synthesis of the other derivatives [15,16]. Currently, DHA, artesunate (ATS), artemether (AM) and arteether (AE; also known as artemotil) are the four derivatives for clinical application [6] (Figure 1). The application of ART and its derivatives in malaria patients is rapid and effective, especially when used alongside other antimalarial drugs in artemisinin-based combination therapies (ACTs) [17,18].

In the early 1990s, the anti-tumor activity of ART-type drugs was discovered [19,20,21], making them also promising candidates for cancer therapy. Since then, their anti-tumor mechanisms have been widely studied in numerous cancer types. It has been proved that tumor cells are more susceptible to ART-type drugs than their normal counterparts, because of their higher intracellular iron capacity and tumor microenvironment [22,23]. When iron cleaves the endoperoxide bridge of ART-type drugs, the generated ROS leads to various cellular events, such as DNA damage, cell cycle arrest and cell death [5,24]. ART-type drug-induced cell death was first addressed as apoptosis. However, more studies demonstrated that ART-type drugs not only induce apoptosis but also other types of regulated cell death (RCD). Here, we provide a timely summary and discussion of research in the last ten years on the influence of ART-type drugs on different RCD pathways in tumor cells (Section 2). On this basis, we review the current combination therapies of ART-type drugs with signaling pathway-related proteins or antibodies to establish enhanced efficacy (Section 3). To overcome the shortcomings of ART-type drugs, such as low solubility, low bioavailability, short plasma half-life and chronic toxicity, ART-related nanocarrier delivery technology came into the picture and proved to be more tumor-specific with improved effectiveness. Therefore, we recap the formation and efficacy of ART-type drugs-loaded nanoparticles (NPs) according to the recently published research (Section 4). We hope this review can serve as a comprehensive overview of the cell death-inducing mechanism and promising combination treatment as well as the newest delivery approaches for ART-type drugs.

## 2. Artemisinin and Regulated Cell Death in Cancer

Artemisinin as a potential anti-tumor drug has been investigated for decades. Considerable effort was made to understand the mechanism of ART-induced cell death, in which, RCD drew major attention. RCD is a lethal process resulting in the elimination of unwanted cells through built-in effects of physiological programs (referred to as programmed cell death), or cell death induced by the intracellular or extracellular stress [25,26]. Among all classified RCD types, ART-type drug-induced cell death has been linked to apoptosis, ferroptosis, autophagy, necroptosis and pyroptosis as reviewed in the following sections.

### 2.1. Apoptosis

Apoptosis is an irreversible RCD with certain biochemical feature changes, such as the caspase cascade activation, DNA fragmentation, and the externalization of internal phosphatidylserine [27,28,29]. Cell shrinkage and pyknosis can be observed in early-stage apoptotic cells, followed by the formation and release of the apoptotic body in the late stage [30].

As of today, the extrinsic and intrinsic pathways are the two main apoptosis pathways, and the perforin/granzyme pathway as a T-cell mediated signal route has also been defined. The extrinsic pathway is initiated through interactions of ligands with transmembrane receptors of the tumor necrosis factor (TNF) superfamily, such as TNF-α/TNFR1, FasL/Fas, TRAIL/DR4 or DR5, VEGI/DR3 and TWEAK/Fn14 [31,32,33,34,35,36,37]. The ligand binds to the receptor to recruit adaptor proteins (FADD, TRADD) to activate the caspase cascade and trigger apoptosis [38,39,40]. As a counterbalance to this process, FLICE-like Inhibitory Protein (c-FLIP) competitively binds to FADD or procaspase-8 to prevent the death-inducing signaling complex (DISC) formation and inhibit apoptosis [41]. The intrinsic pathway is normally initiated by cell stress (ROS/radiation, etc.) or insufficient activation of caspase-8 required for the extrinsic pathway [42]. Due to pro-apoptotic signaling, the mitochondrial membrane is permeabilized, which results in the release of cytochrome C and the formation of the apoptosome, leading to the activation of caspase-9 and the following caspase cascade [43]. The perforin/granzyme pathway is induced by the secretion of perforin and granzymes from multiple immune cells to impose sanctions against infected or transformed cells [44]. By entering the targeted cells through the perforin-formed pores, granzyme B induces caspase-3-related apoptosis, while granzyme A leads to DNA fragmentation [45].

ART-type drug-induced apoptosis was initially reported in human leukemic cells by Efferth et al. in 1996 [46]. Later, other studies indicated that ART-type drugs trigger tumor cells to undergo apoptosis through regulating numerous factors within apoptosis pathways (Table 1; Figure 2, green part I) [47,48]. In the extrinsic pathway, DR5 expression is upregulated by ATS and DHA in colon cancer cell lines [49]. However, Ilamathi et al. found that ATS mainly induces DR4 but not DR5 expression in the hepatocellular carcinoma cell line HepG2, indicating the complex impacts of ART-type drugs on apoptosis pathways [50]. Besides, ART also inhibits the expression of cellular anti-apoptotic c-FLIP [51]. Based on the mentioned effects of ART-type drugs, caspase-8 and caspase-3 are inevitably upregulated to induce apoptosis and DNA damage [52,53].

For the intrinsic apoptosis pathway, it has been shown that the ROS accumulation induced by ART-type drugs directly influences the signal transduction within the pathway. However, the convoluted signaling network of this pathway results in a diversified effect of the ART-type drugs. It was shown that the expression of some pro-apoptotic proteins, such as Bax [54], Noxa [55], Puma [56], Bim and truncated Bid (tBid) [57], is upregulated to promote apoptosis. Furthermore, ART-type drugs also suppress the expression of multiple anti-apoptotic proteins, including Bcl-2 [58], Bcl-xL [59] and Mcl-1 [60]. Because of the altered and imbalanced expression of pro- and anti-apoptotic proteins under treatment, mitochondrial outer membrane permeabilization (MOMP) is induced [61], which leads to the release of cytochrome C [58] and activation of caspase-9 [54].

For the perforin/granzyme pathway, it has been reported that ART-type drugs induce granzyme B expression in γδ T cells which derived from the peripheral blood mononuclear cells and cytotoxic T lymphocytes (CTLs), indicating that ART-type drugs directly act on cancer cells with concomitant effects on immune cells to induce tumor cell to undergo apoptosis [62,63].

Apoptosis induced by ART-type drugs has been studied for 30 years. Their effects on the apoptosis-related proteins have been fully analyzed and their potential in cancer treatment has been investigated. Apart from apoptosis, ART-type drugs have also been shown to induce other RCD pathways in tumor cells, which will be discussed below.

**Table 1 pharmaceutics-14-00395-t001:** List of studies from the last 10 years investigating the effect of ART-type drugs related to apoptosis in cancer cells with information on the model systems used, the ART-type drugs tested and the major findings of the study.

Cell Lines; Cancer	Drugs	Effects	Ref.
HOS, MG-63, U-2 OS, Saos-2; Osteosarcoma	DHA	Cell viability↓; Cell apoptosis↑; G2/M phase arrest↑; Cleaved caspase-3, -8, -9↑; BAX↑; Bcl-2↓; FAS↑; Cyclin D1, B1↓; Cdc25B↓; NF-kB activity↓	2011 [52]
MCF-7, T47D, MDA-MB-231; Breast cancer	ATS	Cell death↑; ROS↑; DFO reduces ROS production and cell death; LC3 puncta↑; LC3-II↑; Cell death rescued by CQ and BafA1	2011 [61]
G-361, A375, LOX; Melanoma	DHA	Cell apoptosis↑; ROS↑; Cell viability↓; DFO reduces ROS production and cell death; Transmembrane potential↓; NOXA↑; CHOP↑; p-P53↑	2012 [55]
T47D; Breast cancer	DHA	Cell viability↓; G0/G1 phase↑; Cell apoptosis↑; tBid↑; Cytochrome C↑; Cleaved caspase-8, -9↑; Bim↑; Bcl-2↓	2013 [57]
Eca109, Ec9706; Esophageal cancer	DHA	Cell viability↓; Cell apoptosis↑; G0/G1 phase↑; Swollen mitochondria↑; Apoptotic body↑; Bcl-2, Bcl-xL↓; Bax↑; Pro-caspase-3↓; Caspase-9↑; Cyclin E↓; CDK2, CDK4↓	2013 [59]
SW1990, BxPC-3, PANC-1; Pancreatic cancer. γδ T cell	DHA	No influence on the ell viability of γδ T; DHA-treated γδ T cell reduces cancer cell viability; Increasing expression of perforin, granzyme B, CD107a, IFN-γ from γδ T cell	2013 [63]
HepG2, Huh-7, LO2; Liver cancer	ART, ATS, DHA	Cell viability↓; Cell apoptosis↑; NAC and zVAD reduce cell death; Chromatin condensation↑; ROS↑; Transmembrane potential↓; Caspase-3, -8, -9 activity↑; Cytochrome C releasing↑; Bax, Bak, Bim↑; Mcl-1↓	2015 [60]
Diverse cell lines	ART and 4 derivatives	Cell death↑; Cell apoptosis↑; Transmembrane potential↓; ROS↑; Intracellular calcium↑; G2/M phase↑; Caspase-3 activity↑; Pro-caspase-3, -9↓; Caspase-9↑; Apaf-1↑; P53, Bax↑, Bcl-2↓	2017 [54]
EJ-138, HTB-9; Bladder cancer	DHA	Cell viability↓; Cell apoptosis↑; Transmembrane potential↓; ROS↑; Caspase-3 activity↑; Bax↑, Bcl-2↓; Cytochrome C↑	2017 [58]
Diverse cell lines	ATS	Cell apoptosis↑; Lipid peroxidation↑; GRP78↑; CHOP↑; PUMA↑; Tumor growth↓	2017 [56]
SK-Hep-1; Liver cancer	DHA	Cell viability↓; Cell apoptosis↑; Cleaved caspase-3, -8, -9↑; Cleaved PARP-1↑; Sp1↓; XIAP↓; p-ERK, p-P38, p-JNK↓	2018 [48]
SK-BR-3, MDA-MB-468, MCF-7; Breast cancer	ATS	Cell viability↓; Cell divisions↓; G1 phase↑; CDK1, CDK4↓; CDC25C↓; Cyclin B, Cyclin D3↓; P21↑; Cell apoptosis↑; Cleaved PARP-1↑; Caspases activation↑; Mitochondrial outer membrane permeability↑; Cytochrome C, SMAC↑; ROS↑	2019 [53]
4T1; Mouse breast cancer	ART	Cell viability↓; Cell apoptosis↑; TGF-β↓; Tumor growth↓; Treg and MDSC expansion↓; CD4+ IFN-γ+ T cells and granzyme B+ cytotoxic T lymphocytes↑	2019 [62]

Arrow “↑” indicates an enhancing effect or upregulation; “↓” indicates a diminishing effect or downregulation; abbreviations used in the table are listed at the end of the manuscript.

### 2.2. Ferroptosis

ART-type drugs not only induce apoptosis but also promote ferroptosis in tumor cells. As a recently defined mechanism of RCD, ferroptosis is characterized by iron accumulation and lipid peroxidation [64]. Morphologically, ferroptosis differs from apoptosis, necrosis, or autophagy, exhibiting small mitochondria with increased membrane density and fewer cristae, and it has a relatively intact cell membrane and normal nucleus [65]. Auto-oxidation or lipoxygenases (LOXs)-mediated oxidation of polyunsaturated fatty acids (PUFAs) leads to the production of lipid ROS and eventually triggers ferroptosis [66]. Within this process, glutathione peroxidase 4 (GPX4) is critical for regulating ferroptosis by reducing toxic lipid hydroperoxides to the corresponding alcohol by oxidizing glutathione (GSH) [67]. Besides, GSH is synthesized from glutamate and cysteine, the intracellular concentration of which is regulated by the amino acid antiporter system x_c_^−^ [68]. Thus, the blockage of system x_c_^−^, the lack of GSH, or the inhibition of GPX4 abolishes the anti-oxidative functions and initiates canonical ferroptosis. Furthermore, the imbalance of the intracellular Fe^3+^ and Fe^2+^, or the accumulation of Fe^2+^ in the labile iron pool can stimulate ROS production and trigger non-canonical ferroptosis [69].

ART is known to kill parasites and cancer cells through the ROS which is generated by the cleavage of its endoperoxide bridge [70,71]. Meanwhile, the accumulation of ROS will trigger ferroptosis. Thus, more evidence was found to prove that ART-type drugs are capable of inducing ferroptosis and influencing ferroptosis-related proteins (Figure 2, brown part II). First of all, more ROS generation was observed upon treatment with ART-type drugs, not only in the cytoplasm but also in mitochondria and on lipid level [72,73]. Second, the expression of SLC7A11 and SLC3A2, which encode system x_c_^−^, are downregulated by ART-type drugs to obstruct the glutamate and cysteine exchange [74]. Simultaneously, less GSH, more oxidized glutathione (Glutathione disulfide; GSSG), and reduced GPX4 are detected in the treated cells, which eliminates the pivotally anti-peroxidant system [75]. Lastly, direct interference with ROS formation or iron accumulation limits the effect of treatment with ART-type drugs. The iron chelator deferoxamine (DFO) can rescue cells from ART-type drug-induced ferroptosis [76]. Moreover, similar effects were observed in the cells treated with ROS scavenger Trolox [76] and N-acetyl-l-cysteine (NAC) [77], the ferroptosis inhibitor liproxstatin-1 (Lip-1) and ferrostatin-1 (Fer-1) [78], the lysosomal inhibitor bafilomycin A1 (BafA1) [79] (Table 2).

As a newly defined RCD type, ferroptosis is extremely relevant with ROS accumulation and iron dyshomeostasis. As we demonstrated above, ART-type drugs contribute ROS and induce ferroptosis in various tumor cells. However, there is still a lack of understanding of the mechanism of ferroptosis, for example, the exact role of lipid ROS in executing ferroptosis [80]. Here, we summarize the effects of ART-type drugs on the ferroptosis pathway (Table 2) and hope to provide potential therapeutic approaches for cancer treatment.

**Table 2 pharmaceutics-14-00395-t002:** List of studies from the last 10 years investigating the effect of ART-type drugs related to ferroptosis in cancer cells with information on the model systems used, the ART-type drugs tested and the major findings of the study.

Cell Lines; Cancer	Drugs	Effects	Ref.
Diverse cell lines	ART, 10 derivatives	Artenimol induced cell death rescued by Fer-1 in CCRF-CEM cell	2015 [81]
Panc-1, COLO357, AsPC-1, BxPC-3; Pancreatic cancer	ATS	ROS↑; Cell death rescued by DFO, trolox and Fer-1	2015 [76]
Head and neck squamous cell carcinoma	DHA	GPX4↓; Ras↓; P53↓; Bcl-2↓; Cell death rescued by DFO	2016 [82]
DAUDI, CA-46; Burkitt’s Lymphoma	ATS	Cell death rescued by DFO, Lip-1 and Fer-1; ATF4↑; CHOP↑; CHAC1↑; Tumor growth↓	2019 [83]
U251, U373; Patient-derived glioma	DHA	Cell death↑; ROS and Malondialdehyde↑; GSH↓; GSSG↑; CHOP↑; HSPA5↑; GPX4↑	2019 [75]
PaTU8988, AsPC-1; Pancreatic cancer	ATS	Cell death rescued by Fer-1; GRP78↑	2019 [84]
HL60, KG1, THP-1; Leukemia	DHA	Cell viability↓; Dysfunction of mitochondria; Mitochondrial ROS↑; Cytoplasm ROS↑; p-AMPK↑; p-mTOR↓; Ferritin heavy chain (FTH)↓; GPX4↓; FTH over-expression prevents DHA-induced ferroptosis; Tumor growth↓	2019 [73]
U87, A172; Glioblastoma	DHA	Cell viability↓; Total ROS and lipid ROS↑; HO-1↑; GPX4↓; Mitochondrial ridges↓; Bilayer membrane density↑; Fer-1 decreases ROS production and inhibits cell death	2020 [72]
MT-2, MT-4, HUT-102; Leukemia	ATS	T-cell growth↓; ROS↑; Cell death rescued by Fer-1; Tumor growth↓	2020 [78]
Diverse cell lines	ART ATS DHA AM	Cell death↑; lipid ROS↑; GSH↓; Cell death rescued by DFO or BafA1	2020 [79]
U2932, SU-DHL2, SU-DHL4, SU-DHL6, 293 T; Lymphoma	ATS	Cell viability↓; Colony formation↓; GPX4↓; FTH-1; ROS and Malondialdehyde↑; Cell death rescued by Fer-1; p-STAT3↓; Tumor growth↓	2021 [85]
Hep3B, PLC/PRF/5, Huh7, HepG2; Primary liver cancer	DHA	Cell viability↓; Lipid ROS and Malondialdehyde↑; Iron content↑; GSH/GSSG↓; GPX4↓; SLC7A11 and SLC3A2↓; CHAC↑; Tumor growth↓; p-PERK and IRE1-α↑; ATF4 and ATF6↑	2021 [74]
NCI-H1299, A549, LTEP-a-2, NCI-H23, NCI-H358; Lung cancer	ART DHA	Cell death↑; Cystine/glutamate transporter (xCT)↓; Cell death rescued by NAC	2021 [77]

Arrow “↑” indicates an enhancing effect or upregulation; “↓” indicates a diminishing effect or downregulation.

### 2.3. Autophagy

The role of autophagy (mainly macroautophagy here) in cancer is fundamentally a double-edged sword that can be a tumor suppressor or tumor protector. Thus, it is essential to understand the effects and mechanisms of ART-type drugs in inducing autophagy and the consequences in cancer cells. Autophagy is a catabolic process that degrades aggregated or long-lived proteins, damaged organelles, inactive pathogens in the lysosome [86]. The extracellular and intracellular stress, such as hypoxia, nutrient deprivation, growth factor depletion, ROS accumulation, and DNA damage, are responsible for the initiation of autophagy [87,88]. Multiple autophagy-related proteins (ATGs) assemble as vital complexes (ULK1 complex, PI3KIII complex, or PI3P binding complex) to initiate the phagophore formation, in which AMP-activated kinase (AMPK) acts as the activator and mammalian target of rapamycin (mTOR) as the suppressor. Meanwhile, Microtubule-associated protein 1A/1B-light chain 3 (LC3) is cleaved by ATG4 into LC3-I, which is further modified by ATG7 and ATG3 to conjugate with the lipid phosphatidylethanolamine (PE) to generate LC3-II (LC3-PE) [89]. Then, LC3-II will be incorporated into the pre-autophagosomal membranes [90,91]. Various autophagy receptors (P62, NBR1, Surf1, NIX/BNIP3L) are recruited to bind with LC3-II to recognize and engulf the targeted cargos to form the autophagosome. Finally, the matured double-membrane autophagosome fuses with the lysosome to form autolysosome and degrade the sequestered contents followed by the release of the vital cellular components [92].

ART-type drugs are proved to induce autophagy by the formation of LC3-puncta or the autophagosome (or acidic vesicular organelles) in numerous cancer cells (Figure 2, blue part III). ART-type drugs not only increase LC3-II expression but also trigger ULK1 phosphorylation and downregulate P62/SQSTM expression (Table 3). Besides, the activation of AMPK and inhibition of mTOR expression also indicates that ART-type drugs induce autophagy through AMPK signaling activation and AKT-mTOR signaling suppression [93,94]. Interestingly, the investigation of the crosstalk of autophagy and apoptosis shows inconclusive results: Jia et al. and Jiang et al. found that the autophagy inhibitors 3-Methyladenine (3MA) or Hydroxychloroquine sulfate (HCQ) promote ART-type drug-induced apoptosis [95,96]; others found that the autophagy inhibitors Chloroquine (CQ), Spautin-1, BafA1, 3MA and the ROS scavenger NAC inhibit both autophagy and apoptosis [85,94,97,98,99]. The drug dosage, duration of treatment, and cell specificity could be responsible for the discrepancy in findings.

### 2.4. Other Types of Regulated Cell Death

Necroptosis and pyroptosis are forms of RCD while sharing similar morphological features compared with necrosis. Both induce plasma membrane pore formation and result in rapid lytic cell death. When cells suffer the endogenous or external blockage of the apoptosis pathway, the TNF-α/TNFR1, IFN/IFNR, or the LPS/TLR signaling transduction will initiate necroptosis [111,112]. Subsequently, it triggers the RIPK1 autophosphorylation and the formation of the RIPK1/RIPK3 complex (known as necrosome) [113]. RIPK3 further promotes the oligomerization and translocation of MLKL to the plasma membrane to form pores (Figure 2, yellow part IV), which releases the cellular contents, the Damage-Associated Molecular Patterns (DAMPs) and the other cytokines [114].

Button et al. found that ATS is capable of inducing RIPK1 expression and MLKL phosphorylation in RT4 schwannoma cells [115] (Table 4). However, the ATS-induced RIPK1 expression was not confirmed in the other cell lines [116,117]. Meanwhile, the RIPK1 inhibitor necrostatin-1 (Nec) partially rescues ATS-induced cell death [78,115,116]. The mentioned results are still insufficient to conclude, more evidence on caspase-8 condition, RIPK1/RIPK3 phosphorylation, and morphological analysis is needed.

Pyroptosis is caspase-dependent cell death stimulated by bacteria, virus invasion, chemotherapy drugs, or toxins [120,121]. It is mainly related to inflammasome activation and inflammatory response. Pyroptosis-related pattern recognition receptors (PRRs) include Toll-like receptors (TLRs), intracellular nucleotide-binding oligomerization domain-like receptors (NLRs), and AIM2-like receptors, which recruit procaspase-1 via ASC to form inflammasome [122]. Then, the active caspase-1 cleaves gasdermin D (GSDMD) into N-terminal fragment (GSDMD-NT) to form pores on the cell membrane and result in pyroptosis [123]. It has also been shown that active caspase-3 can perform the same task [124].

It has been reported that chemotherapy drugs (doxorubicin, cisplatin) induce pyroptosis through caspase-3 dependent Gasdermin E (GSDME) activation [125]. Similarly, DHA was shown to induce caspase-3 cleavage to activate GSDME in breast and esophageal squamous cell cancers (Table 4). With the formation of the GSDME pore, higher levels of lactate dehydrogenase (LDH, cell membrane rupture indicator) and interleukin-1β (IL-1β) were detected in treated cells. Moreover, the knockdown of AIM2 or GSDME partially restores the cell viability in DHA-treated cells [118,119]. ART-type drug-induced pyroptosis is mainly through the activation of caspase-3-induced GSDME-NT pore formation (Figure 2, pink part V). This process intersects with ART-type drug-induced apoptosis through caspase-3.

In summary, ART-type drugs induce multiple RCD pathways to execute cell death in cancer cells. Since apoptosis, ferroptosis, autophagy, pyroptosis, or necroptosis (more evidence needed) have a dynamic connection in signal transduction. ART-type drugs have diverse consequences on the cellular level (Figure 2).

## 3. Combination Treatment of ART-Type Drugs with RCD-Targeting Biologics in Cancer

Artemisinin’s anti-tumor properties and mechanisms have been well established. As FDA-approved anti-malaria treatments, ART-type drugs are economically feasible and rapidly available for cancer patients. Although the application of monotherapy (single medication treatment) is still common in some cancer therapeutics, the combination treatments are more effective, less susceptible to drug resistance with fewer side effects [126,127,128]. Based on the ART-related RCD pathways, the combination of ART-type drugs with recombinant proteins or antibodies targeting the respective RCD pathways promises superior specificity and efficacy in cancer.

As a family member of the TNF superfamily, TRAIL induces the caspase-dependent extrinsic apoptosis pathway via binding with death receptors (DR4 or DR5) [129]. Recombinant TRAIL has been extensively analyzed and shown a selective anti-tumor activity in vitro and clinical trials as a tumor therapeutic [130,131,132]. However, some cancer cells are relatively resistant to TRAIL-induced apoptosis, because of the insufficient expression or mutation of death receptors on the cell surface, overexpression of anti-apoptotic proteins (Bcl-2, c-FLIP, XIAP), or the defects of caspases or essential molecules [133]. To stimulate tumor cells to undergo TRAIL-induced apoptosis, ART-type drugs have become prominent due to their linkage with the apoptosis pathway. ART-type drugs not only induce apoptosis themselves but also manipulate the key proteins of the apoptosis pathway in cancer cells. Therefore, the potency of combining recombinant TRAIL with ART-type drugs was investigated in a variety of cancer cells and in vivo. He et al. first carried out the exploration in human prostate cancer cells, demonstrating that DHA upregulates DR5 expression and that the combination treatment significantly promotes tumor cell death [134]. Later, Kong et al. confirmed the effectiveness of this therapy in BxPC-3 cells subcutaneously xenografted into nude mice [135]. Moreover, Ilamathi et al. observed increased DR4 expression in ART-treated hepatocellular cancer cells, in which the efficacy is enhanced under the inhibition of STAT3 [50]. Recently, our research group found that ART-type drugs increase DR5 expression in a P53-dependent manner in wild-type P53 colon cancer cells and that the combination treatment of ATS/DHA with a DR5-specific TRAIL variant remarkably increases cell death in 3D spheroids [49]. Altogether, the combination of ART-type drugs with TRAIL indicates a synergy and improved effects in cancer treatment, which shows great potential for future clinical trials.

Transferrin (TF) is a crucial glycoprotein to maintain iron homeostasis and transport iron into cells via transferrin receptor (TfR)-mediated endocytosis. As an iron-dependent cell death mechanism, ferroptosis is strongly connected with iron metabolism and dyshomeostasis, in which both TF and TfR play important roles [136,137]. Compared with untransformed cells, overexpression of TfR has been observed in cancer cells [138]. Therefore, the combination treatment of recombinant TF with ART has been established to enhance ROS generation, improve ART-induced ferroptosis, and target tumor cells [138]. In 1995, Lai and Singh first combined DHA with Holo-transferrin as a treatment, exhibiting selective cytotoxicity to lymphoblastic leukemia cells [139]. More studies were performed and showed similar effects in different cancer types [140,141,142,143]. Moreover, multiple ART-transferrin conjugates have been established, demonstrating better tumor selectivity and lower toxicity to normal tissue compared with mono-treatment [144,145,146]. Recently, our group revealed that holo-transferring in combination with DHA induces both apoptosis and ferroptosis in breast cancer cells [147]. Apart from TRAIL and TF, ATS was combined with Rituximab, a commercial antibody specifically targeting CD20 on the surface of B cells. The co-treatment significantly decreased the cell viability of non-Hodgkin’s lymphoma cells [148].

For now, the strategy of ART-type drugs combination with biologics is focused on a few targets in the apoptosis and ferroptosis pathways. An additional example was described by Ji et al., who demonstrated that DHA also upregulates FAS expression in osteosarcoma cells [52], which provides a possibility to combine ART-type drugs with FasL fusion proteins to target cancer cells. Besides, it has been found that recombinant human arginase (rhArg) induces cancer cell apoptosis and autophagy via ROS accumulation [149,150,151]. It will be interesting to investigate the efficacy of the combination treatment of ART-type drugs and rhArg in inducing apoptosis and autophagy in cancer treatment. In summary, the combination of biologics with ART-type drugs shows synergistic effects in cancer treatment, yet a wider range of combinations still waits to be explored. For the thus far tested combination treatments, further in vivo studies and clinical trials are needed to assess their true potential in cancer treatment.

## 4. Delivery of Art-Type Drugs with Nanocarriers

The anti-tumor efficacy of ART-type drugs is limited by poor water solubility and bioavailability, short half-life, and chronic toxicity [152]. In order to enhance their therapeutic effectiveness, multiple nanocarriers have been explored for the delivery of ART-type drugs. Nanocarriers are formed by nanomaterials, aiming for the transportation of substances. Progress in ART-loaded nanocarriers has opened up massive opportunities in the recent ten years (Table 5).

Here, we summarize the use of spherical ART-type drug-loaded nanocarriers, which are up to several hundred nanometers in size. Due to their physicochemical and biological properties, they have the potential to be taken up more easily by cells, including the transportation across the blood-brain barrier (BBB) [196,197]. The tumor environment is more acidic than the normal cell environment, due to the altered cell metabolism, hypoxia and deficient blood perfusion [198]. According to this feature, pH-responsive nanoparticles have been developed enabling the specific release of encapsulated drugs in tumor tissues [175]. Moreover, nanocarriers can facilitate the co-delivery of multiple therapeutics (small molecules and biologics) and cofactors, such as ions. This improves tumor targeting and anti-tumor efficacy, and reduces the occurrence of drug resistance compared to monotherapy [199].

For ART-type drugs, several pH-sensitive nanoparticles and co-delivery systems have been developed and tested [162,167,200]. The latter often rely on the fact that ART-type drugs are radical precursors which are activated by metal ions such as Fe^2+^, Mn^2+^, and Ni^2+^ through cleavage of the endoperoxide bridge by a Fenton-like reaction. Meanwhile, co-delivery systems with RCD-targeting biologics and small molecules have been explored.

In this section, we discuss different formulations of nanocarriers loaded with ART-type drugs. Information is grouped by nanocarrier material (inorganic-based nanoparticles, liposomes, polymer-based nanoparticles, carbon-based nanoparticles, and other types of nanocarriers) [197].

### 4.1. Inorganic ART-Loaded Nanoparticles

Inorganic-based nanoparticles are typically composed of inorganic compounds, such as alumina, silica, metals, and metal oxides. Compared to organic materials, they are non-toxic, hydrophilic, biocompatible and highly stable [162]. The size, shape and porosity of the nanoparticles are determined by the selected compounds used in the inorganic-based synthesis [201,202].

#### 4.1.1. Ion-Containing Inorganic ART-Loaded Nanoparticles

The unique magnetic properties of ion-containing inorganic nanoparticles make them highly efficient for imaging and diagnosis in cancer therapy [162]. Moreover, since ART-type drugs are considered radical precursors, ion-containing inorganic nanoparticles have attracted great attention for cancer nanotherapy [203]. Zhang et al. developed visible light-sensitive iron oxide nanoparticles (IONPs): Hyaluronic Acid (HA)-TiO_2_-IONPs/ART (~40 nm size), which realized the co-delivery of ART and Fe^2+^. The in vitro data showed HA-TiO_2_-IONPs generates ROS upon irradiation with visual light, resulting in apoptosis in breast cancer cell line MCF-7. Meanwhile, tumor-bearing BALB/c mice treated with HA-TiO_2_-IONPs did not show apparent body weight loss, which demonstrated promising safety profiles. Moreover, the plasma half-life of HA-TiO_2_-IONPs is 2.85-fold higher than that of ART [156]. ART derivatives also showed promising efficacy when capsulated into metal ion-based inorganic nanoparticles [154,161].

Other metal oxides are also employed as nanoparticle materials. ART-loaded pH-responsive mesoporous NiO (mNiO-Tb-ART) nanoparticles were engineered to release Ni^2+^ at acidic pH, which reacts with the endoperoxide bridge in ART to produce radicals [204]. Chen et al. developed Fe_3_O_4_@MnSiO_3_-FA which supplies Mn^2+^ and ART to the tumor in A549 xenograft mouse models. According to cell viability assays, they demonstrated that Mn^2+^ is more effective in catalyzing the Fenton-like reaction than Fe^2+^. This was the first report to demonstrate that Mn^2+^ assists ART to achieve higher anticancer efficiency [153]. Nanoparticles constituted by hollow mesoporous manganese trioxide (Mn_2_O_3_) and ART (TKD@RBCm-Mn_2_O_3_-ART) were shown to have a drug release percentage of 97.5%. Furthermore, the release of Mn_2_O_3_ in mice enhanced the magnetic resonance imaging (MRI) capability and can be used for the diagnosis of breast cancer. [162]. Zn^2+^-Fe^2+^ co-released (ZnFe_2_O_4_) NPs showed promising bio-imaging capacity and anti-proliferative effect in HeLa, A375 and HepG-2 cell lines [158].

#### 4.1.2. Other Inorganic ART-Loaded Nanoparticles

Other inorganic-based nanoparticle studies also improved the anti-cancer effects. Due to the combination of silicon hydrophilic features with promising stability, Luo et al. developed mesoporous silica nanoparticles co-loaded with ART, buthionine-sulfoximine, and transferrin peptide. TF peptide guides the nanoparticle to TfR overexpressing malignant tumor cells and cathepsin B control the release of ART and buthionine-sulfoximine to increase ROS production, decrease GSH levels and thus induce tumor cell death [160]. In another design, dual-metal-organic-frameworks (MOFs)@ART release ART due to the pH-responsive degradation of outer MOFs in an acidic tumor environment. Strong absorbance in the near-infrared region by inner MOFs makes the combination treatment of chemo-photothermal therapy promising. Furthermore, dual MOFs improved imaging by MRI and fluorescence optical imaging (FOI) [155].

### 4.2. ART-Type Drug-Loaded Liposomes

Liposomes, the first developed drug carriers, are simple sphere-shaped vesicles with a diameter of 80–300 nm, consisting of one or more phospholipid bilayers. As they are directly taken up by cells through endocytosis or fusion with cell membranes, they can enhance the absorption of the encapsulated drugs [196,205,206,207]. The encapsulation by liposomes also helps to increase the stability and reduce the toxicity of the agent. They are non-toxic, biocompatible, completely biodegradable, and non-immunogenic. However, they are limited by low solubility, unexpected leakage of encapsulated drugs, and high production costs [205].

Single ART and ART derivatives loaded in liposomes were shown to have anti-tumor efficacy [164,172]. Building on that, pH-responsive release mechanisms were explored to increase the accumulation of ART-type drugs specifically at tumor sites, thereby reducing side effects [163,167,176]. For example, Dadgar et al. combined ART, phosphatidylcholine and polyethylene glycol 2000 together into a liposomal formulation, in which polyethylene glycol increases the solubility and the encapsulation of ART. This nanoliposomal formulation showed better cytotoxic ability than standard ART in the MCF-7 cell line [163].

Furthermore, co-delivery of ART-type drugs with cisplatin resulted in greatly enhanced anti-cancer effects to A549 cells in vitro. And in vivo, co-delivery of ART-type drugs with sorafenib, epirubicin, or daunorubicin was demonstrated to have synergistic anti-cancer effects in liver tumors, non-small cell lung cancer (NSCLC), and breast cancer [169,170,171,173,174]. Invasive brain glioma was efficiently treated by co-delivery of paclitaxel and artemether in mannose-vitamin E derivative conjugate (MAN-TPGS1000) and DQA-PEG2000-DSPE liposomes since these liposomes can easily penetrate the BBB [165]. The co-delivery of TF and ART-type drugs in liposomes also resulted in targeted drug delivery and enhanced therapeutic efficacy in liver and breast tumor-bearing mouse models [166,175].

Lastly, metal ions were also applied in liposomes. Fu et al. loaded ART into the inner space of hollow mesoporous silica (HMS) liposomes, with Fe_3_O_4_ immobilized on the surface. ART@HMS-Fe_3_O_4_ is taken up into lysosomes and in the acidic lysosomal environment, Fe^2+^ is released to activate ART, which leads to cytotoxicity in ductal carcinoma ZR75-30 cell line [168].

### 4.3. Polymer-Based ART-Loaded Nanoparticles

Polymeric nanoparticles can be defined as solid particles made up of polymers with a size range of 1–100 nm. Not only chemicals but also protein or DNA material can be carried by polymeric nanoparticles [208]. They are widely used because of the high loading capacity, easy preparation technique, controllable and uniform size distribution, and longer clearance time, which is advantageous in delivering the drugs to a particular site at a particular rate [208].

An in vitro study indicated that using a dendritic-linear-dendritic hybrid copolymer based on hyperbranched 2,2-bis (hydroxymethyl) propionic acid (bis-MPA) and linear PEG chains to deliver ART-type drugs leads to anticancer biological effects on breast cancer cell lines MCF-7 and MDA-MB-231 [190]. Furthermore, GEL/DHA and hyaluronan /DHA nanoparticles inhibited the proliferation of NSCLC A549 cells [181]. Moreover, an in vivo study showed that ATS-loaded M-HFn nanoparticles accumulate on the tumor tissue and efficiently inhibit tumor growth in 4T1-bearing mice [186]. Most importantly, several co-delivery systems based on polymeric nanoparticles were established: first, co-delivery of ART-type drugs and doxorubicin (DOX) or paclitaxel were demonstrated with superior results compared to free ARTs, which could be explained by the enhanced permeability and the retention effect [182,183]. Second, co-delivery of ART-type drugs with TF enhanced their tumor inhibition ability. For example, the pH-responsive DHA-GO-TF and TF-8arm-PEG-DHA delivery systems both showed higher solubility, enhanced tumor delivery specificity, minimal side-effects, and superior tumor growth restrained ability in vivo [184,185]. Third, the release of metal ions by polymer-based nanoparticles was shown to enhance ART or ART derivative-induced tumor inhibition [187,188,189].

### 4.4. Carbon-Based ART-Loaded Nanoparticles

The discovery of fullerene (C60), carbon nanotube, and graphene introduce carbon nanomaterials to become hotspots as novel nanostructures. Since its discovery, C60 has attracted increasing interest in practical applications, and the biological activity, including antiviral, antioxidant, and chemotactic activities were tested. Moreover, C60 could generate cytotoxic ROS under laser exposure [209]. Due to these unique physicochemical properties, fullerene and graphene oxide were employed to load ART derivatives in nanoparticle delivery systems. The porous structure of carbon nanoparticles provides a large surface area, resulting in plentiful exposed modifiable sites [210]. The surface modification, such as the binding of functional groups or molecules, increases the feasibility of biological application [211]. For example, with the modification of HA, HA-fullerene (C60) showed water-soluble properties with improved dispersion ability in physiological environments than C60. C60 lacks electrons, making it a good electron acceptor to interact with TF. So, the co-delivery system HA-C60-Tf/ATS was designed with excellent biocompatibility and enhanced antitumor efficacy in S180 tumor mouse models [191].

### 4.5. Other Types of ART-Type Drugs-Loaded Nanocarriers

Liposomes are widely used due to their biodegradable and non-toxic properties. However, the unexpected leakage and uncontrolled fast release of liposomes also limited their applications. Thus, solid lipid nanoparticles (SLN) have been made with solid lipids to protect loaded drugs from harsh environments and achieve sustained drug release [212,213,214]. However, the solubility of most drugs in solid lipids is not that high, which restrained the loading capacity of SLNs. Therefore, the nanostructured lipid carriers (NLCs) were designed to incorporate liquid lipid into the solid matrix. Compared with SLNs, NLCs have higher loading capacity, slower polymorphic transition, and low crystallinity index [212]. The TF-ART-NLCs were made by Compritol^®^, Tween 80, oleic acid, and dichloromethane, and their cytotoxicity to the U87MG cell line was far greater than free ART [192].

Micelles are composed of lipid monolayers, where hydrophobic and hydrophilic components assemble into nano-sized spheres. They have a normal size ranging from 5 to 100 nm, depending on the types of the hydrophilic components and length of hydrophobic chains. Micelles have the ability to increase drug solubility, reduce toxicity, and prolong circulation time making them a suitable model for drug delivery systems [215]. ART was encapsulated inside of poly (E-caprolactone)-poly (ethylene glycol)-poly (E-caprolactone) (PCL-PEG-PCL) micelles, biotin-PEG-PCL micelles, or lymphatics-homing peptide (LyP-1) conjugated PEG-PCL micelles. They all significantly increased the drug accumulation in tumors, leading to a better tumor growth inhibition capacity than free ART [177,178,179].

Niosomes share a similar structure with liposomes, but liposomes are made up of phospholipids with two hydrophobic tails, while niosomes are non-ionic surfactant vesicles with a single hydrophobic tail. Similar to micelles, niosomes are able to capture both hydrophilic and hydrophobic drug molecules, while being more stable than liposomes. Therefore, several ART-type drugs were encapsulated with the niosome technique [194,205]. Asgharkhani et al. prepared the ART-loaded niosomes by a certain ratio of Span-60:Tween-60:PEG-600:ART, showing better inhibitory effects to MCF-7 cell line (one-fourth IC_50_ compared to free ART) [193]. Compared to free artemether, artemether-loaded nano-niosomes induced more tumor necrosis and inhibited tumor growth in a 4T1-bearing mouse model [195]. Artemisone, a 10-amino-artemisinin derivative of ART, was packed into niosomes to test the inhibitory effects on the melanoma A-375 cell line. In vitro data showed that the niosome formulation of artemisone displayed high cytotoxicity to melanoma cells but no toxicity to normal skin cells. This study also packed artemisone into lipid-based nanoparticles, which showed similar results [194].

Multiple nanotechnologies have been applied to ART-type drugs and have shown different therapeutic efficacy. In general, compared with free ART-type drugs, these nanocarriers have improved bioavailability and stability. The different drug release mechanisms and co-delivery systems help to reduce side effects and improve their anti-tumor effect. In recent years, ion-containing inorganic ART-loaded nanoparticles have attracted increasing attention. Although these ART-type drug-based nanotechnologies show promising anti-tumor efficacy in vivo, the clinical data is not abundant. Further evaluation is needed.

## 5. Summary

ART-type drugs were developed to treat malaria patients and became one of the most effective drugs ever since. However, their anti-tumor activity has also attracted extensive attention due to their promising therapeutic effects in vivo and in vitro. This review systematically summarized the effect of ART-type drugs on RCD pathways and discussed the approaches to enhance their efficacy. The combination treatment with biologics, and the nanoparticles delivery methods improve the lethality and specificity of ART-type drugs for tumor cells. Despite the promising results thus far, it is important to also investigate new targets for the combination treatment based on the known RCD pathways. Furthermore, for now, all studies were performed on cultured cells or animal models. The lack of clinical trials makes the curative effect unpredictable and more in-depth studies are necessary.

## Figures and Tables

**Figure 1 pharmaceutics-14-00395-f001:**
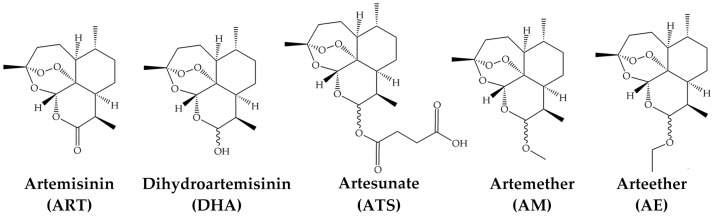
Artemisinin and four derivatives in clinical application.

**Figure 2 pharmaceutics-14-00395-f002:**
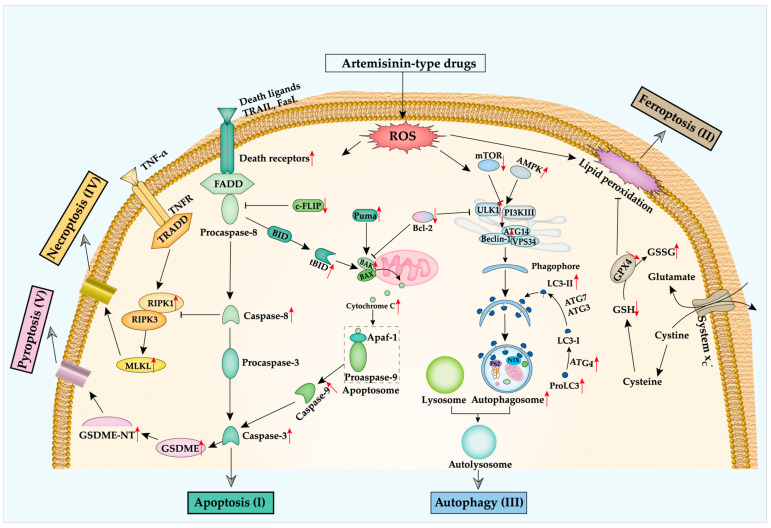
Artemisinin-type drugs influence several RCD pathways. The red arrows represent the upregulation or downregulation of certain key players on protein level. (**I**) ART-type drugs produce ROS resulting in increased expression of death receptors, and cleavage of caspase-3 and caspase-8, and decreased c-FLIP expression, leading to stimulation of the extrinsic apoptosis pathway. Meanwhile, ROS itself also triggers the intrinsic apoptosis pathway by upregulating the expression of tBID, BAX/BAK, inducing the release of Cytochrome C to activate caspase-9. (**II**) ROS accumulation from ART-type drugs results in lipid peroxidation to induce ferroptosis. Besides, the disruption of the oxidative homeostasis maintaining system by ART-type drugs leads to downregulation of GPX4 expression. (**III**) The oxidative stress from ART-type drugs initiates the formation of the phagophore via activating the AMPK pathway and assembling the VPS34 complex, resulting in autophagy. ART-type drugs also increase MLKL pore formation to induce necroptosis (**IV**), or GSDME-NT pore formation to induce pyroptosis (**V**).

**Table 3 pharmaceutics-14-00395-t003:** List of studies from the last 10 years investigating the effect of ART-type drugs related to autophagy in cancer cells with information on the model systems used, the ART-type drugs tested and the major findings of the study.

Cell Lines; Cancer	Drugs	Effects	Ref.
K562; Leukemia	DHA	Autophagosome formation↑; LC3-I and LC3-II↑; ROS↑; TfR↓; Cell viability↓	2012 [100]
Eca109, Ec9706; Esophageal cancer	DHA	Autophagosome formation↑; LC3-I and LC3-II↑	2013 [59]
Diverse cell lines	DHA	Autophagosome and autolysosome formation↑; LC3-I and LC3-II↑; P62↓; p-IκBα; ROS scavenger 4-Hydroxy-TEMPO (TEMPO) reduces autophagic vacuoles	2014 [101]
BxPC-3, PANC-1; Pancreatic cancer	DHA	Cell growth↓; LC3-1↓; LC3-II↑; 3MA enhances DHA-induced apoptosis; p-JNK↑; Beclin 1↑; ROS↑; JNK inhibitor and beclin-1 siRNA suppress DHA-induced autophagy	2014 [95]
Cal-27; Tongue squamous cell carcinoma	DHA	Cell viability↓; Colony formation↓; Autolysosome formation↑; LC3-II↑; DNA damage↑; Nuclear p-STAT3↓; Beclin-1↑; Tumor growth↓	2017 [102]
SKOV3; Ovarian cancer	ATS DHA	Cell viability↓; Beclin-1↑; LC3-II↑; Autophagosome formation↑; Cell viability rescued by CQ and BafA1	2018 [98]
Cholangiocarcinoma	DHA	Cell viability↓; Colony formation↓; LC3-I and LC3-II↑; P62↓; PI3KC1↓; AKT and mTOR↓; BCL-1↓; Vps34↑; Beclin-1↑; Spautin-1 inhibits DHA-induced autophagy and cell death	2018 [99]
Diverse cell lines	DHA-37	Cell viability↓; Cell viability rescued by autophagy inhibitors CQ, 3-MA or LY294002; LC3-II↑; P62↓; Autolysosome formation↑; HMGB1↑; p-MAPK and P38↑; Tumor growth↓	2018 [103]
HCT116; Colon cancer	ATS	Cell viability↓; Autolysosome formation↑; Atg5↑; Beclin-1↑; LC3-II↑; Autophagy inhibitor HCQ promotes ATS-induced apoptosis; Tumor growth↓	2018 [96]
SU-DHL-4, SU-DHL-10, OCI-LY3; Diffuse large B cell lymphoma	SM1044	Autolysosome formation↑; LC3-II↑; Autophagy inhibitors CQ and BafA1 inhibit DHA-induced apoptosis; p-AMPK↑; ULK1↑; Ceramide↑; Caramide inhibitor S1P and l-cycloserine, the Ca^2+^/calmodulin-dependent kinase kinases inhibitor STO-609 inhibit AMPK activation; Tumor growth↓	2018 [97]
HepG2215; Hepatocellular carcinoma	DHA	Cell viability↓; Colony formation↓; DNA damage↑; Autolysosome formation↑; P62↓; LC3-II↑; ROS↑; cell mobility↓;	2019 [104]
HeLa; Cervical cancer	DHA	Cell viability↓; Tumor growth↓; LC3 puncta↑; LC3-II↑; Autolysosome formation↑; ROS↑; γH2AX↑; DNA damage↑; p-mTOR	2019 [105]
Eca109; Esophagus squamous cell carcinoma	DHA	Cell viability↓; Tumor growth↓; ROS↑; LC3 puncta↑; P62↓; LC3-II↑; TRF2↓; NAC reduces LC3 puncta	2020 [106]
Diverse cell lines	DHA	Cell viability↓; Colony formation↓; Tumor growth↓; LC3-II↑; Beclin-1↑; P62↓; Autolysosome formation↑; IFI16↓; Ra1B↓; USP33↓	2020 [107]
TE-1, Eca109; Esophageal cancer	DHA	Cell migration↓; LC3 puncta↑; LC3↑; P62/SQSTM↓; 3MA or overexpression of Akt restores DHA-suppressed migration; p-AKT and p-mTOR↓; E-cadherin↑; N-cadherin↓; Vimentin↓	2020 [108]
EJ, T24; Bladder cancer	ATS	Cell viability↓; Cell migration↓; Colony formation↓; Autolysosome formation↑; p-AMPK and p-ULK1↑; p-mTOR↓; LC3-II/I ratio↑; 3MA inhibits ATS-induced apoptosis; AMPK activator enhances ATS-induced autophagy and apoptosis; AMPK inhibitor, 3MA, and NAC suppresses ATS-induced apoptosis; ROS↑	2020 [94]
BON-1, QGP-1; Pancreatic neuroendocrine cancer	ATS	Cell viability↓; Cell death rescued by 3MA; LC3-II↑; DHA induces apoptosis, ferroptosis, and autophagy	2020 [109]
Ishikawa, AN3CA; Endometrial carcinoma	ATS	Cell viability↓; Cell migration↓; CD155↑; P62↓; LC3-II/I ratio↑; ATG5↑; ATS-treated cancer cell triggers NK92 cytotoxicity	2021 [110]
U2932, SU-DHL2, SU-DHL4, SU-DHL6, 293 T; Diffuse large B cell lymphoma	ATS	Cell viability↓; Colony formation↓; Apoptosis↑; P62↓; LC3-II/I ratio↑; Acidic vesicular organelles formation↑; CQ reduces ATS-induced apoptosis; p-STAT3↑; Knockdown of STAT3 enhances ATS-induced autophagy, apoptosis, and ferroptosis; Tumor growth↓	2021 [85]

Arrow “↑” indicates an enhancing effect or upregulation; “↓” indicates a diminishing effect or downregulation.

**Table 4 pharmaceutics-14-00395-t004:** List of studies from the last 10 years investigating the effect of ART-type drugs related to necroptosis and pyroptosis in cancer cells with information on the model systems used, the ART-type drugs tested and the major findings of the study.

Cell Lines; Cancer	Drugs	Effects	Ref.
**Necroptosis**			
Diverse cell lines	ATS	Cell viability↓; p-MLKL↑; RIPK1↑; Caspase inhibitor z-VAD-fmk (zVAD), Nec and siRIPK1 rescue ATS-induced cell death	2014 [115]
Diverse cell lines	ATS	Cell viability↓; ROS↑; Mitochondrial ROS↑; zVAD, Nec, siRIPK1, and ROS scavengers rescue ATS-induced cell death;	2017 [116]
MT-2, MT-4, HUT-102; Leukemia	ATS	T-cell growth↓; ROS↑; Nec rescues ATS-induced cell death; Tumor growth↓	2020 [78]
**Pyroptosis**			
MCF-7, MDA-MB-231; Breast cancer	DHA	Cell viability↓; Colony formation↓; LDH↑; AIM2↑; Cleaved caspase 3↑; GSDME/DFNA5↑; HMFB1↑; IL-1β↑; shAIM2 and shDFNA5 restore cell survival and colony formation; Tumor growth↓	2021 [118]
Eca109, Ec9706; Esophageal squamous cell carcinoma	DHA	Cell viability↓; LDH↑; IL-1β↑; GSDME-NT↑; Cleaved caspase 3↑; Caspase inhibitor Ac-DEVD-CHO reduces GSDME-NT, LDH, IL-1β, and rescue cell viability; Tumor growth↓	2021 [119]

Arrow “↑” indicates an enhancing effect or upregulation; “↓” indicates a diminishing effect or downregulation.

**Table 5 pharmaceutics-14-00395-t005:** List of studies reporting the delivery of ART-type drugs with nanocarriers detailing the carrier material, cargo, model systems and main findings of the studies compared to free drugs.

Carrier Materials	Cargo	Cell Lines; Cancer	Main Outcomes	Ref.
**Inorganic-based NPs**				
MnSiO_3_, Fe_3_O_4_	ART	A549; Lung cancer	Mn^2+^ release↑; Antitumor activity in vivo↑	2015 [153]
Fe (III) carboxylate	DHA	HeLa; Cervical cancer. A549; Lung cancer	Co-release of DHA and Fe^3+^; Complete tumor cure with no observable side effects on normal tissues	2016 [154]
Dual metal-organic-frameworks	ART	HeLa; Cervical cancer	High tumor inhibition rate (~2-fold of free ART); No obvious effect on the major organs of mice	2016 [155]
HA-TiO_2_	ART	MCF-7; Breast cancer	Generation of ROS under visual light irradiation; Higher concentration of ART in tumor tissue	2017 [156]
Mesoporous NiO, Tb-DPTA	ART	HeLa; Cervical cancer	Ni^2+^ release↑; Antitumor activity in vitro and in vivo↑	2018 [157]
ZnFe_2_O_4_	ART	Diverse cell lines	Lower cell viability than free ART	2018 [158]
SiO_2_, Fe_3_O_4_	ART	HepG-2; Liver cancer	Easy release of Fe^2+^ by weak acidic etching; Enhanced production of ROS with NIR light irradiation	2019 [159]
Mesoporous silica	ART, TF	MCF-7; Breast cancer. CT26; Colon cancer	Co-delivery of iron to cancer cells; Release of ART in the presence of cathepsin B; ROS↑; Glutathione↓; Anti-cancer efficacy in vitro and in vivo↑	2019 [160]
FeCl_3_ · 6H_2_O, Na_3_Cit · 2H_2_O, NaOAc	DHA	MCF-7, MDA-MB-231, MDA-MB-453; Breast cancer	Fe^2+^ release↑; High toxicity to intractable breast cancer cells	2020 [161]
Hollow mesoporous manganese trioxide	ART, Mn	MCF-7; Breast cancer	Deep penetration of solid tumors	2021 [162]
**Liposomes**				
PPC, PEG2000	ART	MCF-7; Breast cancer	Half IC_50_ compared to free ART	2013 [163]
P90G, CHOL	DHA	MCF-7; Breast cancer	Better cellular uptake efficiency	2014 [164]
DQA-PEG2000-DSPE	AM, DOX	C6; Brain cancer	Transport of drug across BBB, elimination of brain CSCs; Destruction of vasculogenic mimicry channels	2014 [165]
DPPC, DSPC, CHOL	ART, TF	MCF-7, MDA-MB-231; Breast cancer	10- and 5.5-fold higher levels of ART and TF production than free drugs; Tumor volume in mice↓	2015 [166]
DPPC, mPEG2000	ART Dimer	MDA-MB-231; Breast cancer	Better anti-tumor efficacy than Paclitaxel	2015 [167]
Hollow mesoporous silica, Fe_3_O_4_	ART	ZR75-30; Ductal carcinoma	Lysosomal environment-responsively released ART result in decreased cell viability	2017 [168]
EPC, CHOL, PEG2000-DSPE	DHA, Epirubicin	MDA-MB-435S, MDA-MB-231, MCF-7; Breast cancer	Drug circulation↑; Targeting delivery to the tumor; Anticancer efficacy↑ than free DHA or Epitubicin	2018 [169]
DSPE-PEG2000-NHS	DHA, Daunorubicin	MDA-MB-435S; Breast cancer	More accumulation in tumor than free DHA; Better antitumor efficacy with no obvious toxicity in mice	2018 [170]
CHOL, cRGD-PEG-DSPE, phospholipids, Fe_3_O_4_	ART, Cisplatin	A549/R; NSCLC	The 15.17-fold lower IC_50_ value of free cisplatin against A549/R cells, ROS↑; Cell apoptosis rates↑	2018 [171]
FeCl_3_ · 6H_2_O, FeSO_4_ · 7H_2_O, sodium oleate, sodium hydroxide, Acetonitrile.	DHA	HNSCC; Head and neck squamous cell carcinoma	Significant targeting effect in a magnetic field; Better inhibition of HNSCC in mice than free DHA	2019 [172]
Cholesteryl oleate, glyceryl trioleate, CHOL, DOPE	DHA, SRF	HepG2; Liver cancer	BAX and Bcl-2↑; Exhibited a 3-fold higher SubG1% of cells than free DHA or SRF	2019 [173]
EPC, CHOL, DSPE-PEG2000, DSPE-PEG2000-R8	DHA, Epirubicin	A549; NSCLC:	Increased drug accumulation; Enhanced specificity and anti-tumor efficacy in vivo	2019 [174]
DSPE-PEG2000, DOPE, CHEMS	DHA, TF	HepG2; Liver cancer	High oxidative state at the tumor site; Eradication of HepG2 tumor in mice	2020 [175]
DSPE-PEG2000-HE-R6	ART	4 T1; Breast cancer	Longer retention time in tumors and higher efficiency in tumor suppression	2021 [176]
**Micelles**				
PEG-PCL	ART	MDA-MB-435S; Breast cancer	Specific delivery of ART to tumor site; Higher antitumor efficacy than other ART formulations in vivo with low toxicity	2012 [177]
PCL-PEG-PCL	ART	MCF-7, 4T1; Breast cancer	Prolong in vivo residence time in rats	2018 [178]
Biotin-PEG-PCL	ART	MCF-7; Breast cancer	Tumor inhibition; No toxic effects on HFF2 fibroblast cells	2019 [179]
**Polymer-based NPs**				
mPEG	ATS	L1210; Leukemia. MCF7; Breast cancer	Controllable release of ATS in the presence of esterase	2014 [180]
Formulation I: Gelatin; Formulation II: Hyaluronan	DHA	A549; NSCLC	Formation of DHA nanosized aggregates in an electrostatic field; Higher anticancer proliferation activities than DHA alone in A549 cells.	2014 [181]
Poly(lactic-*co*-glycolic acid) and 1,2-dipalmitoyl-sn-glycero-3-phosphocholine	DHA, DOX	HeLa; Cervical cancer. HepG2; Liver cancer	Increased doxorubicin accumulation in cell nuclei; cytotoxicity↑	2015 [182]
PEG	DHA, Paclitaxel	HT-29; Colon cancer	Higher accumulation in the tumor site; Tumor growth in vivo↓; Systemic toxicity↓	2015 [183]
Graphene oxide	DHA, TF	EMT6; Breast cancer	Significant enhancement of delivery specificity and tumor cytotoxicity; Complete tumor cure in mice	2015 [184]
PEG	DHA, TF	LLC; Lung cancer	High solubility (~102-fold of free DHA); Relatively high drug loading; Circulating half-life↑; One-fifth the size of the tumor in free DHA	2016 [185]
H-apoferritin	AS	Hela; Cervical cancer	pH-responsive release of AS; Cytotoxic ROS↑; Cytotoxicity↑; Biocompatibility↑; No additional side effects	2019 [186]
PNE, FeOOH	ART	4T1; Mouse breast cancer	Extremely low toxicity to normal tissue; Tumor elimination after 7-day treatment; No tumor recurrence in 30 days after treatment.	2019 [187]
Iron coordinated hollow polydopamine nanospheres	DHA	HeLa; Cervical cancer	3.05-fold higher anti-tumor efficacy than free DHA	2019 [188]
PEOz-PLA-PBAE	ATS dimer	CT-26; Colon cancer	Enhanced cellular uptake of the drug depot by the cancer cells; Enhanced anti-tumor efficacy in vivo	2020 [189]
Bis-MPA, PEG	ART	MCF-7, MDA-231; Breast cancer	Completely non-toxic towards healthy fibroblasts	2021 [190]
**Carbon-based NPs**				
HA-C60	AS	MCF-7; Breast cancer	Increased intracellular accumulation of AS in tumor; Remarkably enhanced antitumor efficacy	2015 [191]
**NLCs**				
Cholesterol, oleic acid, stearylamine	ART	U87MG; Malignant gliomas	High entrapment efficiency; Controlled drug release for brain administration	2018 [192]
**Niosomes**				
Span 60, Tween 60, PEG-600	ART	MCF-7; Breast cancer	4-fold higher cytotoxic activity than free ART	2014 [193]
Span 60, CHOL	Artemisone	A-375; Melanoma	Highly selective cytotoxicity towards melanoma cells, not to normal skin cells	2015 [194]
Span, Tween, CHOL	AM, Paclitaxel	4T1; Mouse breast cancer	Superior tumor necrosis and smaller tumor volume than free AM	2020 [195]

Arrow “↑” indicates an enhancing effect or upregulation; “↓” indicates a diminishing effect or downregulation.

## Data Availability

Not applicable.

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
