# Peer review of "Artemisinin-Type Drugs in Tumor Cell Death: Mechanisms, Combination Treatment with Biologics and Nanoparticle Delivery"

_pharmaceutics, 2022, doi:10.3390/pharmaceutics14020395_

Round 1

Reviewer 1 Report

This is a well-written review manuscript describing the molecular mechanisms of artemisinin and its derivatives on a variety of cell death. The review is comprehensive and well organized. a minor comment is that the selectivity  of these compounds on normal cells vs. cancer cells and direct molecular targets of these compounds related to cell death mechanisms are not well discussed, if it is possible.

Author Response

This is a well-written review manuscript describing the molecular mechanisms of artemisinin and its derivatives on a variety of cell death. The review is comprehensive and well organized. a minor comment is that the selectivity of these compounds on normal cells vs. cancer cells and direct molecular targets of these compounds related to cell death mechanisms are not well discussed, if it is possible.

Response: We really appreciate this affirmation of our manuscript. In agreement with the reviewer’s comment, we added more description in Line 59-63 to explain “the selectivity of these compounds on normal cells vs. cancer cells” and a summary sentence that these compounds release ROS to induce multiple cellular events. As a chemically reactive compound in the presence of Fe2+, artemisinin exerts broad effects on a lot of proteins involved in cell death, so we summarized its published effects and possible targets in section 2.

Reviewer 2 Report

The authors present a narrative review paper concerning the use of artemisinin-type drugs in cancer and their delivery using nanosystems. Although the article is well organized and outlined; I think there are some aspects that the authors need to clarify and improve. 

Authors should include a list of abbreviations in tables and figures. Tables and figures should be self-explanatory without the need to read the main text of the manuscript.

In section 1. Introduction, line 60, the authors state that: “Here, we provide … in the latest ten years on the influence of ART-type drugs to different RCD pathways in tumor cells (section 2).”. This is not true, since in some mechanisms, for example apoptosis (section 2.1), the authors cite studies from 1996 (reference 41), from 2020 (reference 42 and 44) and from 2012 (reference 50).

When the authors explain the various mechanisms of action of ART-type drugs throughout section 2 they should refer to figure 2 for a better understanding of the processes involved in these mechanisms.

Why didn't the authors include a table of the studies that investigated the effect of ART-type drugs related to apoptosis (section 2.1.), as they did with the other mechanisms?

Some steps in the mechanisms of action are not well explored due to the complexity of the topic. For example, it is not very clear what γδ T cells are? What mTOR means? And rhArg?  

Authors need to include references in some statements:

  • “ART is known to kill …..of its endoperoxide bridge.” Lines 149 – 150
  • “Recombinat TRAILS has been…. a selective anti-tumor activity in vitro and in clinical trials as a tumor therapeutics.” Lines 262 – 264. The authors should include references related to in vitro as well as clinical trials.
  • “This can improve tumor targeting and anti-tumor efficacy, and reduce the occurrence of drug resistance compared to monotherapy.” Lines 326 – 327
  • “For ART-type drugs, several pH-sensitive nanoparticles and co-delivery systems have been developed and tested. The latter often…. Fenton-like reaction. However, also co-delivery…been explored” Lines 328 – 3332 Authors should provide the references of such studies.

I suggest using the term “nanosystem” which I consider broader than the term “nanoparticle”, since it includes all types of nanoparticles and other nanosystems such as liposomes that are vesicles, micelles, nanotubes.

Authors should present an official definition of nanosystems.

Authors should be careful with the expressions / terminology used. They should revise the term "ART nanoparticles". This may suggest that ART is used as a matrix for the preparation of the nanoparticles. The authors should use the term "ART-loaded nanoparticles”.

The information that the authors present in Table 4 should be supplemented to have some utility for the readers. For example, authors could include the composition of the nanosystems, the physical-chemical properties that justify the advantages of their use, and the main outcomes (for example, when compared to the free drug).

Authors should explain this statement:

“The lipid-based nanoparticles have a similar design approach as liposomes, but are made up by a monolayer of phospholipids, …”

This statement is not correct. Liposomes are a lipid-based nanosystems. The authors meant to refer to solid lipid nanoparticles (SLN) or nanostructured lipid carriers (NLC)? So, this part “…made up by a monolayer of phospholipids” is not totally correct.

I suggest that the authors investigate what is each of the presented nanosystems.

Author Response

We thank the reviewer for the comments regarding our manuscript. The suggestions raised have been helpful for further improving the quality and the clarity of this manuscript (especially for section 4). Our responses (in bold) to the comments (in plain text) are given below.

Point 1: Authors should include a list of abbreviations in tables and figures. Tables and figures should be self-explanatory without the need to read the main text of the manuscript.

Response 1: Thank you for your nice suggestions. We have a long list of abbreviations for this manuscript, and after discussion with the editor, we now add the list of abbreviations at the end of the paper immediately after the paragraph "Conflict of Interest". Besides, the explanatory text of the tables and figures has been added.

Point 2: In section 1. Introduction, line 60, the authors state that: “Here, we provide … in the latest ten years on the influence of ART-type drugs to different RCD pathways in tumor cells (section 2).”. This is not true, since in some mechanisms, for example apoptosis (section 2.1), the authors cite studies from 1996 (reference 41), from 2020 (reference 42 and 44) and from 2012 (reference 50).

Response 2: We only use the older references to introduce the history of apoptosis and artemisinin research. We started to introduce the mechanisms of artemisinin evoking cell death from Line 114, and all the cited references on the mechanisms summarized in all the tables are published after 2011. The exact publishing year can be found in the Ref. column in each table.

Point 3: When the authors explain the various mechanisms of action of ART-type drugs throughout section 2 they should refer to figure 2 for a better understanding of the processes involved in these mechanisms.

Response 3: Thank you for the nice suggestion, we referred each section to Figure 2. In line 114, 164, 208, 231, and 260, respectively.

Point 4: Why didn't the authors include a table of the studies that investigated the effect of ART-type drugs related to apoptosis (section 2.1.), as they did with the other mechanisms?

Response 4: We didn’t add the table of the effect of ART-type drugs related to apoptosis previously, because there are several reviews that summarized it. While considering the reviewer’s suggestion, we add a new Table 1 to summarize the effects of ART-type drugs on apoptosis in cancer cells in Line 140.

Point 5: Some steps in the mechanisms of action are not well explored due to the complexity of the topic. For example, it is not very clear what γδ T cells are? What mTOR means? And rhArg?  

Response 5: Thank you for your suggestion to make the manuscript more intelligible. γδ T cells are a type of T cells derived from the peripheral blood mononuclear cells. The description has been added in line 132.

mTOR is short for the mammalian target of rapamycin, which is a central regulator of cellular and organismal growth and homeostasis. The full name has been added in line 198.

rhArg is short for recombinant human arginase, which has been developed for arginine deprivation therapy for cancer treatment. It has been reported that rhArg also produces ROS in cancer treatment. We added more descriptions in lines 330-334 to make it clear. Besides, the full name of each abbreviation has been listed at the end of the manuscript.

Point 6: Authors need to include references in some statements:

  • “ART is known to kill …..of its endoperoxide bridge.” Lines 149 – 150
  • “Recombinant TRAIL has been…. a selective anti-tumor activity in vitro and in clinical trials as a tumor therapeutics.” Lines 262 – 264. The authors should include references related to in vitro as well as clinical trials.
  • “This can improve tumor targeting and anti-tumor efficacy, and reduce the occurrence of drug resistance compared to monotherapy.” Lines 326 – 327
  • “For ART-type drugs, several pH-sensitive nanoparticles and co-delivery systems have been developed and tested. The latter often…. Fenton-like reaction. However, also co-delivery…been explored” Lines 328 – 3332 Authors should provide the references of such studies.

Response 6: References were added to each mentioned location.

Line 162: “ART is known to kill …..of its endoperoxide bridge.”

Line 288: “Recombinant TRAIL has been…. a selective anti-tumor activity…”

Line 357: “This can improve tumor targeting and anti-tumor efficacy…”

Line 359: “For ART-type drugs, several pH-sensitive nanoparticles and co-delivery…”

Point 7: I suggest using the term “nanosystem” which I consider broader than the term “nanoparticle”, since it includes all types of nanoparticles and other nanosystems such as liposomes that are vesicles, micelles, nanotubes.

Authors should present an official definition of nanosystems.

Response 7: Thank you for your comprehensive consideration. By searching the literature and consulting Prof. Anna Salvati, a specialist in our institute, we introduced nanocarrier as a broader term in our manuscript. The use of nanoparticles and nanocarriers has been adjusted accordingly in the manuscript.

Point 8: Authors should be careful with the expressions / terminology used. They should revise the term "ART nanoparticles". This may suggest that ART is used as a matrix for the preparation of the nanoparticles. The authors should use the term "ART-loaded nanoparticles”.

Response 8: Thanks for the suggestion. This indeed can lead to confusion. We replaced ART nanoparticles with "ART-loaded nanoparticles”.

Point 9: The information that the authors present in Table 4 should be supplemented to have some utility for the readers. For example, authors could include the composition of the nanosystems, the physical-chemical properties that justify the advantages of their use, and the main outcomes (for example, when compared to the free drug).

Response 9: In response to this suggestion, we now add the carrier materials and main outcomes of the ART-loaded nanoparticles in Table 5.

Point 10: Authors should explain this statement:

“The lipid-based nanoparticles have a similar design approach as liposomes, but are made up by a monolayer of phospholipids, …”

This statement is not correct. Liposomes are a lipid-based nanosystems. The authors meant to refer to solid lipid nanoparticles (SLN) or nanostructured lipid carriers (NLC)? So, this part “…made up by a monolayer of phospholipids” is not totally correct.

Response 10: We meant the nanostructured lipid carriers (NLC). For a better understanding, we now add the composition of the NLC in lines 487-489.

Point 11: I suggest that the authors investigate what is each of the presented nanosystems.

Response 11: Based on your suggestion, we checked more literature about each nanocarrier and added more descriptions for better understanding in section 4.

Round 2

Reviewer 2 Report

The narrative review paper has become far more appealing.

Final suggestions:

  • Abbreviations should be listed alphabetically.
  • The authors should review again the concepts of nanosystems, namely liposomes, SLN and NLC. What defines NLC is not that they: "...are made up of a monolayer of phospholipids". This is not correct. NLCs may not even have phospholipids in their composition.

Author Response

We thank the reviewer for the comments regarding our manuscript. Our responses (in bold) to the comments (in plain text) are given below.

  • Abbreviations should be listed alphabetically.

Response: Thanks for the suggestion. We reorganized the abbreviations alphabetically.

  • The authors should review again the concepts of nanosystems, namely liposomes, SLN and NLC. What defines NLC is not that they: "...are made up of a monolayer of phospholipids". This is not correct. NLCs may not even have phospholipids in their composition.

Response: Thanks for the nice suggestion to improve our manuscript. Based on the comments, we now correct the definition of NLCs and add more descriptions about SLNs and NLCs in Lines 487-498.